# The Role of the IL-6 Cytokine Family in Epithelial–Mesenchymal Plasticity in Cancer Progression

**DOI:** 10.3390/ijms22158334

**Published:** 2021-08-03

**Authors:** Andrea Abaurrea, Angela M. Araujo, Maria M. Caffarel

**Affiliations:** 1Breast Cancer Group, Oncology Area, Biodonostia Health Research Institute, 20014 San Sebastian, Spain; andrea.abaurrea@biodonostia.org (A.A.); angela.araujo@biodonostia.org (A.M.A.); 2IKERBASQUE, Basque Foundation for Science, 48009 Bilbao, Spain

**Keywords:** cancer, epithelial–mesenchymal transition, epithelial–mesenchymal plasticity, cytokines, IL-6, oncostatin M (OSM), invasion, migration

## Abstract

Epithelial–mesenchymal plasticity (EMP) plays critical roles during embryonic development, wound repair, fibrosis, inflammation and cancer. During cancer progression, EMP results in heterogeneous and dynamic populations of cells with mixed epithelial and mesenchymal characteristics, which are required for local invasion and metastatic dissemination. Cancer development is associated with an inflammatory microenvironment characterized by the accumulation of multiple immune cells and pro-inflammatory mediators, such as cytokines and chemokines. Cytokines from the interleukin 6 (IL-6) family play fundamental roles in mediating tumour-promoting inflammation within the tumour microenvironment, and have been associated with chronic inflammation, autoimmunity, infectious diseases and cancer, where some members often act as diagnostic or prognostic biomarkers. All IL-6 family members signal through the Janus kinase (JAK)–signal transducer and activator of transcription (STAT) pathway and are able to activate a wide array of signalling pathways and transcription factors. In general, IL-6 cytokines activate EMP processes, fostering the acquisition of mesenchymal features in cancer cells. However, this effect may be highly context dependent. This review will summarise all the relevant literature related to all members of the IL-6 family and EMP, although it is mainly focused on IL-6 and oncostatin M (OSM), the family members that have been more extensively studied.

## 1. Introduction

Epithelial–mesenchymal transition (EMT) is a dynamic process in which epithelial cells reorganize their cytoskeleton, lose apical–basal polarity, cell–cell adhesions and acquire mesenchymal phenotypes with increased migration and invasion capacities [1]. In the opposite process, mesenchymal–epithelial transition (MET), motile, spindle-shaped mesenchymal-like cells reorganize their cytoskeleton, resulting in an organized epithelium [1]. Both EMT and MET are key processes during physiological (embryonic development, regeneration) and pathological processes, such as cancer and fibrosis [1].

Epithelial–mesenchymal transition (EMT) has been a controversial concept in the last years. The criteria to define EMT have been based on context-specific or even research-community-specific observations, leading to discrepancies in data interpretation and persistent disagreements about whether the process studied in vivo could be considered EMT or not [2,3,4,5,6,7,8,9]. Besides, increasing evidence supports that the complexity, plasticity and diversity of EMT manifestations have been underestimated or oversimplified, and that cells can adopt various degrees of mixed epithelial and mesenchymal features, leading to a continuum of intermediate hybrid phenotypes known as “partial” EMT, which is a heterogeneous, dynamic and reversible process [10,11,12]. Those partial EMT states have been reported to be more relevant to cancer metastasis than the mesenchymal phenotype itself [13], and multiple partial EMT states have been proposed and verified to coexist in the same tumour. Moreover, it was recently suggested that the number of potential intermediate EMT states within a tumour can be an indicator of malignancy [14].

Therefore, the EMT International Association (TEMTIA) has recently published a consensus statement to avoid misinterpretations of the generated research data and encourage scientists to adhere to the term EMP (epithelial–mesenchymal plasticity) instead of the traditional EMT [1]. Therefore, in this work, the term EMP will be used even if the cited authors considered it EMT in their original work.

Epithelial–mesenchymal plasticity (EMP) plays critical physiological roles during embryonic development, postnatal growth and epithelial homeostasis, but it is also involved in a number of pathological conditions, including wound repair, fibrosis, inflammation and cancer. EMP has been intimately linked with most, if not all, of the steps during cancer development and progression (e.g., migration, invasion, immune escape, drug resistance and metastatic dissemination), and these links have been extensively reviewed [15,16,17,18,19,20,21].

During cancer progression, tumour cells and the surrounding tumour microenvironment (TME) components (e.g., immune cells, fibroblasts, endothelial cells) secrete a variety of inflammatory mediators (including cytokines, chemokines, matrix metalloproteinases (MMPs)) that can foster the acquisition of mesenchymal features in cancer cells by fuelling the EMP-promoting inflammation [22,23,24]. Tumour microenvironment-associated inflammation, mainly regulated by cytokines, has been long described to contribute to every stage of cancer progression [25,26,27,28]. Pro-inflammatory cytokines alert the immune system to potential threats, playing a fundamental role in the body’s immune defence. However, dysregulated cytokine production can result in harmful responses to the body and consequent development of pathological conditions, including cancer [29].

The interleukin 6 (IL-6) cytokine family plays important physiological roles in inflammation, immune responses and haematopoiesis [30]. Deregulation of IL-6 signalling has been associated with chronic inflammation, autoimmunity, infectious diseases and cancer, where it often acts as a diagnostic or prognostic indicator of disease activity and response to therapy [31,32]. IL-6 has been considered a keystone cytokine in the link between inflammation and cancer, and has received a lot of attention as a potential therapeutic target [33]. 

The IL-6 cytokine family includes IL-6, IL-11, IL-27, IL-31, ciliary neurotrophic factor (CNTF), leukaemia inhibitory factor (LIF), cardiotrophin 1 (CT-1), cardiotrophin-like cytokine (CLC) and oncostatin M (OSM), all sharing the common glycoprotein 130 (gp130, IL6ST) receptor signalling subunit. Cytokine specificity is dependent on its unique cell-surface receptor, which shows a restricted expression pattern and dimerises with the ubiquitously expressed gp130 subunit. This restricted pattern is what makes some cells more responsive to certain cytokines than others [31]. All IL-6-related cytokine receptor complexes transduce intracellular signals via the Janus kinase (JAK)–signal transducer and activator of transcription (STAT) pathway, but they can also activate the mitogen-activated protein kinase (MAPK)–extracellular signal-regulated kinase (MAPK/ERK) and phosphoinositide 3-kinase (PI3K)–protein kinase B (PKB or Akt) pathways [34]. Activated STATs enter the nucleus and activate context dependent genes [35]. STAT3 signalling is considered a major pathway for cancer inflammation due to its frequent activation in malignant cells and key role in regulating many crucial genes associated with inflammation in the tumour microenvironment [36]. STATs also induce the suppressors of cytokine signalling (SOCS) [37,38], which bind to tyrosine-phosphorylated JAK and tyrosine-phosphorylated gp130 [39], respectively, to stop IL-6 family signalling by means of a negative feedback loop. Interestingly, the IL-6 family has been described to have a great deal of promiscuity, with some members presenting affinity to more than one specific receptor and being able to induce the same or different physiological outcomes [31]. 

This work aims to summarize the role of the IL-6 cytokine family on epithelial–mesenchymal plasticity during cancer. This review is especially focused on IL-6 and OSM, the most studied members in the family, which are generally considered pro-tumorigenic and strong EMP promoters. Information regarding the implication of LIF, IL-11, IL-30 and IL-27 on EMP is also summarized at the end. To our knowledge, there is no reported information linking the remaining members of the IL-6 family with EMP, letting them out of the scope of this review.

## 2. Role of IL-6 and OSM in Epithelial–Mesenchymal Plasticity

The reciprocal causative link between the EMP and IL-6 family has been supported by a large number of independent studies. The experimental evidence linking OSM and EMP is less, but it is still supported by a considerable number of scientific reports. The main effects of IL-6 and OSM in EMP and the signalling pathways involved are summarised in Figure 1. In general, IL-6 and OSM induce EMP in cancer cells by favouring the acquisition of mesenchymal traits and cancer stem cell (CSC)-associated features. They also promote stemness, migration, invasion and metastasis in different in vitro and in vivo experimental settings of several cancer types, including breast, head and neck, lung and gastric cancers.

### 2.1. Signalling Pathways Mediating IL-6 and OSM-Induced EMP

The STAT3 transcription factor is the leading EMP effector upon activation of the IL-6 receptor (IL-6R) and OSM receptor (OSMR) [40,41], although the involvement of other downstream effectors cannot be disregarded. Indeed, it has been recently described that STAT3 phosphorylation at Ser727 and Tyr705 differentially regulates the EMT–MET switch and cancer metastasis [42]. In brief, this study showed that disseminated cancer cells underwent MET by the LIFR/p-ERK/pS727-STAT3 signalling pathway in early stages of lung cancer metastasis, acquiring enhanced sphere-forming ability and proliferative potential. At later stages, cells underwent EMP and acquired CSC characteristics by the activation of the IL6R/pY705-STAT3 signalling pathway. 

In breast cancer, the imperative role of STAT3 as the mediator of EMP induction by IL-6 family is well accepted [43]. Furthermore, it has been shown that tropomyosin receptor kinase C (TrkC) can activate the JAK2/STAT3/Twist1 axis directly and indirectly through Src, leading to IL-6 overexpression and activation of a positive feedback loop to induce and maintain an EMP phenotype [44]. Besides, in head and neck squamous cell carcinoma (HNSCC) cells, IL-6 has been shown to activate the Akt, Erk1/2 and STAT3 pathways, but EMP promotion was demonstrated to be mediated specifically by the JAK/STAT3 pathway [45]. However, the P70S6K kinase was identified by another group as a partial mediator of IL-6 induced EMP in HNSCC, and it could be activated by any of the JAK/STAT3, PI3K/Akt/mTOR and MAPK/ERK pathways downstream of IL-6 signalling [46].

Recently, knockdown of OSMR expression in gastric cancer cells was shown to significantly inhibit cell proliferation, migration, invasion and EMP in vitro, as well as tumorigenesis and metastasis in vivo induced by OSM/STAT3/focal adhesion kinase (FAK)/proto-oncogene tyrosine-protein kinase Src (Src) signalling [47]. However, OSM-induced stemness and the CSC-like phenotype have been both attributed not only to STAT3 but also to PI3K kinase activation [48,49,50]. In non-transformed prostate cells, OSM, but not IL-6, was shown to induce EMP markers, increase migration in 2D and promote the pro-invasive dissemination of cells out of 3D spheroids. Interestingly, among all the effects induced by OSM, only the pro-invasive properties of the spheroids were shown to be STAT3 dependent [51]. 

STAT3 constitutes an integrative node downstream of several hormones, growth factors and cytokine receptors, among other direct STAT3 activators. Therefore, a plethora of complex interactions exist between the IL-6 cytokine family and other classical pathways inducing EMP and stemness, such as the epidermal growth factor receptor (EGFR) [52,53], insulin-like growth factor receptor 1 (IGF1R) [54,55,56], IL-8 [57], Wnt/β-catenin [58] or transforming growth factor-β (TGF-β)/SMAD pathways. Cooperative STAT3/SMAD complexes have been shown to be required for IL-6-, OSM- and TGF-β-induced EMP, invasiveness and stemness in breast, lung and biliary cancers, with a strong synergy between both the JAK/STAT3 and TGF-β /SMAD pathways [43,49,50,59,60]. Moreover, a recent study demonstrated that IFN-β impairs EMP by interfering with the STAT3/SMAD3/Snail axis that links OSM and TGF-β signalling [61]. Nevertheless, the long non-coding RNA HNF1A-AS1 has been shown to target OSM and inhibit EMP, tumorigenesis and metastasis of gastroenteropancreatic neuroendocrine neoplasms via TGF-β signalling [62]. This work is focused only on the effect of the IL-6 cytokine family on EMP, mostly mediated by STAT3, with a brief summary of the implication of other signalling pathways in this section. For a deep comprehension of the implication of JAK/STAT signalling in the regulation of metastasis, CSCs and chemoresistance by EMP, other exhaustive reviews can be considered [40,41].

Apart from membrane receptor-mediated signalling, induction of EMP by the IL-6 family can be influenced by context-specific environmental factors, such as hypoxia [63], or oncogenic activation of transcription factors, such as NFKB [64,65,66]. Furthermore, the c-Myc cofactor PIM1 was identified as an IL-6/STAT3 downstream mediator of the acquisition of EMT and CSC-associated features in cancer cells, suggesting a role for c-Myc on IL-6/OSM-promoted EMP [67]. In fact, OSM has been shown to induce senescence in non-transformed human mammary cells by STAT3/SMAD3-mediated c-MYC suppression, while constitutive c-Myc expression dismantled OSM/STAT3/SMAD3-induced senescence and cooperated with OSM to drive EMT and invasiveness [49,50]. Therefore, c-Myc is considered a “molecular switch” that enables OSM to collaborate with TGF-β in driving the tumorigenicity of breast cancer cells [68].

Although in most situations OSM can be considered a potent EMP inducer, it has been described to promote anti-EMP and anti-metastatic effects in certain circumstances, mainly mediated by STAT1. Opposite to the pro-EMP role of STAT3, STAT1 activation by OSM has been shown to reduce metastasis of lung adenocarcinoma by STAT1/HDAC1/PIAS4-mediated Slug repression [69]. Moreover, OSM was shown to prevent TGF-β from induction of FoxC2 on kidney cells, as well as a variety of matricellular proteins (e.g., SPARC, thrombospondin, CTGF, tenascin C) known to enhance the functional output of EMP programs partly by STAT1 [70]. In non-transformed human kidney cells, OSM was shown to phosphorylate STAT1/3 and ERK1/2/5, leading to pro- and anti-EMP functions. OSM-mediated N-cadherin repression was shown to be mediated by ERK1/2, while ERK1/2 inhibition did not affect E-cadherin repression [71].

In summary, there is strong experimental evidence supporting the role of the transcription factor STAT3 as the main mediator of EMP induction by IL-6 and OSM in cancer [40,41,43,47]. STAT3 can cooperate with other signalling nodules (mainly Src, SMAD and c-Myc) to mediate IL-6 and OSM promotion of EMP in cancer cells [43,44,47,49,50,59,60,67,68]. Interestingly, a few reports have described anti-EMP and anti-metastatic effects of OSM mediated by STAT1 [69,70,71]. As will be described in further sections, it is widely accepted that activation of STAT3 by IL-6 related cytokines leads to EMP while activation of STAT1 leads to anti-tumorigenic and anti-EMP effects.

### 2.2. Activation of Epithelial-Mesenchymal Transition Transcription Factors by IL-6 and OSM

Transcription factors involved in EMP (EMT-TFs) have been recently classified as core and non-core EMT-TFs [1]. EMT-TFs that can be activated by the IL-6 family cytokines are summarized in Table 1, Table 2 and Table 3. IL-6 is able to induce the expression of EMT-TFs in a wide variety of cancer types. IL-6 was reported to induce Snail and Twist in a panel of oestrogen receptor-positive (ER+) human breast cancer cells in vitro, while only Twist levels were increased in xenografts generated by MCF7 cells ectopically expressing IL-6 [72]. Additionally, adipocyte-derived IL-6 has been shown to promote EMP by increasing Zeb1, Snail and Twist1 expression both in ER+ and negative (ER-) cell lines. Of note, only the Twist and Snail levels were decreased after IL-6 blockade [73].

In colorectal cancer (CRC) cells, IL-6 treatment increases the Snail, Slug, Zeb1, TCF3, Fra-1 and FoxQ1 levels [84,85], while in oesophageal, bladder and pancreatic (PDAC) cancer cell lines, different combinations of Snail, Slug, Zeb1, Zeb2, Twist1 and Twist2 transcription factors have been observed to be activated by IL-6 [87,90,91]. In PDAC cells, IL-6 also induces the expression of the Nrf2 transcription factor [87]. However, only Snail has been documented to be increased by IL-6 in lung and head and neck carcinoma cell lines [45,79,82]. Recently, IL-6 has also been shown to induce EMP by direct STAT3 binding to the JunB promoter in uveal melanoma cells [93].

Furthermore, IL-6 can also modulate EMT-TFs at post-translational levels. In breast cancer, IL-6 activates deubiquitinase 3 (Dub3), leading to Snail, Slug and Twist1 protein stabilization [75,114], and it has been shown to stabilize the Twist protein levels through the activation of casein kinase 2 (CK2) in HNSCC [83]. Interestingly, the induction of EMT-TFs can also foster an increased expression of IL-6 via Twist [72], Snail [115,116] or Zeb1 [117], supporting the existence of positive feedback loops. 

It is widely accepted that Snail is increased after OSM exposure in many cancer types, including prostate [51], pancreatic [95], cervical [96] and breast cancers [43,48,61,97]. In addition, OSM has been described to increase the Zeb1 protein levels in PDAC [95] and breast cancer [97] and Zeb2 in cervical cancer [96]. 

As described in the previous section, OSM can inhibit EMP via STAT1, and these effects were concomitant with Slug reduction in lung adenocarcinoma cells [69].

### 2.3. EMP Regulation by IL-6 and OSM through Micro-RNAs (miRNAs)

Apart from the transcriptional regulation, the EMT-TF levels can be altered by a variety of post-transcriptional, translational and post-translational regulatory networks, including micro-RNAs (miRNAs). The list of miRNAs directly or indirectly associated with EMP is extensively increasing [118]. So far, the miR200 [119,120] and miR34 [121,122] families are the best characterized, both being miRNA families and strong protectors of the epithelial phenotype in different cellular systems and positively regulated by the tumour suppressor p53 [121,123,124]. As we will describe next, both miRNA families have been shown to be modulated by IL-6 and OSM in cancer cells. 

IL-6 can activate an IL-6R/STAT3/miR-34a feedback loop in colorectal, breast and prostate cancer cells and it has been demonstrated that miR-34 repression is required for IL-6-induced EMT and invasion [86]. On the other hand, copy number gain of the T-cell malignancy 1 (MCT1) gene has been found to promote EMP, cancer stemness and M2 macrophage polarization in triple-negative breast cancer (TNBC) by direct suppression of miR-34a and consequent IL-6/IL-6R increased signalling, reinforcing the reciprocal signalling between IL-6 and miR-34 [125]. Additionally, IL-6/miR-506-3p/FoxQ1 [85] and IL-6/miR-33a/Twist [92] axes have been identified as the underlying mechanisms of IL-6-mediated EMP. Moreover, IL-6 has been linked to Lin28/Let7 signalling, where NFKB-activated Lin28 inhibits Let7, leading to increased IL-6 levels and consequent NFKB activation, enforcing a positive feedback loop that contributes to the generation and maintenance of CSC-like populations in breast cancer [64].

Not surprisingly, OSM has also been shown to induce EMP by regulating the Lin28/Let7/HMGA2 and miR200/Zeb1 circuits via STAT3 in breast cancer, leading to the initiation and maintenance of an EMP program both in vitro and in vivo [97].

### 2.4. Molecular and Functional Implications of IL-6 and OSM-Induced EMP in Cancer Cells

Most of the studies cited in the previous sections reported EMP using complementary observations, including morphologic, functional or behavioural changes in cancer cells, along with regulation of EMP factors at the protein and mRNA levels. This is actually fundamental since the expression of EMT-TFs alone does not imply that EMP is occurring [1]. Many EMP studies have been focused on breast cancer, and an important proportion of them support a strong link between the EMP process and the generation and maintenance of cancer stem cell-like (CSC-like) cells and therapeutic resistance [126,127,128,129,130,131], in addition to the well-known association of EMP with cell migration and invasion and metastasis. In this section of the review, we will summarise the functional implications of IL-6 (Table 1) and OSM (Table 2) induction of EMP in cancer cells.

An inverse correlation between IL-6 and E-cadherin expression has long been reported [72,132]. In oesophageal and breast cancer cells, IL-6 exposure has been shown to promote mesenchymal/fibroblastic morphology, migration and invasion in vitro and in vivo, associated with reduced CD24, KRT19, EpCAM and E-cadherin expression and increased vimentin, N-cadherin and MMP9 levels [57,72,73,74,76,90]. Similarly, IL-6 treatment increased migration in uveal melanoma cells, together with reduced expression of cell adhesion molecules (TJP1, TJP2 and CDH1), and increased levels of focal adhesion molecules (FN1 and ICAM-1) and fibronectin receptor integrin subunits [93]. Interestingly, the induction of EMP and invasiveness by IL-6/STAT3 has been shown to be mediated by Fra-1 expression in CRC [84] and by PTTG1 in prostate cancer [133]. Moreover, a positive IL-6/STAT3/RTVP-1 feedback loop was shown to be responsible of increased mesenchymal transformation, invasiveness and resistance to anti-tumour treatments in glioma cells [134].

In cervical, HNSCC and non-small cell lung cancer (NSCLC) cells, IL-6 induces a similar pro-mesenchymal morphology, cell scattering, motility and invasion, and decreases E-cadherin and increases N-cadherin and vimentin levels in vitro [94] and in xenografts in nude mice [45,79]. Of interest, these effects were more pronounced in the CD133+ stem-like cells in the lung [80]. Additionally, the IL-6/COX-2/PGE2 axis has been shown to increase the translocation of cytoplasmic β-catenin to the nucleus, leading to EMP and increased invasion in NSCLC cells [81]. 

It is well accepted that IL-6 promotion of EMP is associated with stemness, which includes increased CD133 expression, a CD44 high/CD24 low phenotype, an increased self-renewal or a multilineage differentiation capacity, accompanied by any of the above-mentioned EMP-associated classical alterations. These phenotypic changes have been reported in many cancer types, such as pancreatic, lung, breast and oesophageal cancers [57,59,76,77,78,80,88,90,135].

Multiple studies have demonstrated that IL-6-mediated EMP and stemness provide resistance to conventional chemo(radio)therapy in different cancer types [59,88,90,136] and even to targeted therapy, such as trastuzumab [77] or tyrosine kinase inhibitors (TKIs) [137,138]. However, a recent study has reported that CAF-derived IL-6 mediates EMP but that this effect is not responsible for the increased chemoresistance observed in bladder cancer cells [91]. Although this study is contradictory to previous results, there is solid experimental evidence supporting that IL-6 mediated EMP promotes resistance to therapies in cancer [59,77,88,90,136,137,138].

Anoikis is a form of programmed cell death that occurs in anchorage-dependent cells when they detach from the surrounding extracellular matrix (ECM). Resistance to anoikis has been linked to EMP, stemness and metastasis [1]. IL-6 has been shown to confer anoikis resistance in melanoma, ovarian, pancreatic and cervical cancer cells [139,140,141,142].

Regarding OSM, our group previously characterized that, in cervical squamous cell carcinoma cells with high OSMR levels, OSM induces a mesenchymal morphology, reduces E-cadherin levels and cell cohesion, and increases fibronectin, MMP9, MMP10 and vimentin levels [96]. It also induced tumour sphere formation and lung colonisation of cervical cancer cells in nude mice. Similar OSM-induced phenotypic changes have been documented in breast cancer, where OSM transformed cobblestone-shaped epithelial breast cancer cells into fibroblast-like cells with elongated morphology, multiple protrusions and reduced intercellular adhesion by shifting E-cadherin and β-catenin localization from membranous to cytoplasmic [43,48,97,98].

In breast cancer, OSM also reduces the E-cadherin levels and increases the levels of mesenchymal markers. As it happens for IL-6, there is some controversy about the mesenchymal markers activated by OSM, depending on the cellular context and the experimental setting, mainly the duration and the concentration of the cytokine stimuli. Increased fibronectin and vimentin and reduced E-cadherin protein levels have been documented in MCF-7 and T47D cells after 6 days of OSM exposure, together with a prominent perinuclear aggregation of fibronectin and changes in cell shape. This EMT-like process was accompanied by the induction of a variety of pro-invasive and pro-metastatic genes (e.g., matrix metalloproteinases MMP1, 2, 7 and 9, COX-2, VEGF and CXCR4), and a strong increase in migration and invasion in vitro and metastasis in vivo [97]. However, a shorter 48 h-exposure to OSM was also able to induce the mesenchymal protein S100A7 in both cell lines, while E-cadherin, fibronectin and N-cadherin were only altered in MCF7 cells, and no changes on vimentin levels were observed in any of the two cell lines [48]. In contrast with the two aforementioned studies [48,97], OSM derived from cancer associated adipose tissue was shown to mediate EMP and reduce the E-cadherin levels, without inducing changes on vimentin, fibronectin or cytokeratin expression [98]. Of note, lower concentrations of OSM were used in this study.

As mentioned before, c-Myc is aberrantly expressed in many breast cancers [143]. Its cooperation with OSM has shown to increase vimentin and reduce the E-cadherin protein levels concomitant with increased cellular outgrowths, resulting on cell detachment and migration from 3D in vitro spheroids [49]. Moreover, in an 84-gene “ECM and adhesion molecules profiler array”, 40% of the genes were upregulated by OSM together with many ECM remodelling proteases (MMP1, 2, 3, 7, 8, 9, 10, 11, 12, 13, 16) [49].

Regarding the effects of OSM in mediating stemness and pluripotency, OSM was shown to induce a mesenchymal CD44 high/CD24 low phenotype in breast and pancreatic cancer cells, and this action was unique to OSM and not observed following IL-6 exposure in pancreatic cancer cells [48,95,99]. However, OSM-induced CD44 was shown to contribute to breast cancer metastatic potential through cell detachment but not EMP itself in another study [144]. Moreover, a recent report studied a collection of 28 cytokines and growth factors with altered mRNA expression in the TME of invasive human breast carcinomas and associated with relevant clinical parameters, such as tumour recurrence, metastasis or patient mortality. This study found that six of the nine cytokines that promoted the expansion of mesenchymal/CSC-like cells were members of the IL-6 cytokine family (i.e., IL-6 itself, OSM, LIF, CNTF, CT-1 and CTL), with OSM providing the most potent effect. OSM exposure increased CD44 and Snail expression and repressed CD24, Claudin-1 and E-cadherin in breast cancer cells, while promoting migration and 3D tumour sphere formation [43]. 

In glioblastoma, overexpression of the OSM receptor (OSMR) by ANXA2 led to activated STAT3 and enhanced cell invasion, angiogenesis, proliferation and mesenchymal plasticity [145]. Interestingly, hypoxia activated the ANXA2–STAT3–OSMR signalling axis [145]. Moreover, it has been recently published that mesenchymal-like transitions in glioblastoma cells are driven by macrophage-derived OSM [146].

Overall, most of the findings reported in this section point to similar molecular and functional consequences of IL-6 and OSM-induced EMP in cancer cells. Both IL-6 and OSM induce E-cadherin repression, increase the expression of mesenchymal markers, promote mesenchymal/fibroblastic morphology, stemness and migration, as well as invasion in vitro and in vivo. In addition, IL-6 has been reported to induce resistance to cancer therapies and anoikis. Although most of the evidence is derived from studies performed in cancer cell lines, the pro-EMT and pro-invasive effects of IL-6 and OSM have been proved in vivo in animal models of breast, lung, gastric, cervical and head and neck cancers, among others.

### 2.5. Association between OSM, IL-6 and EMP in Cancer Patient Samples

Despite some controversy regarding the direct effect of EMP on cancer metastasis, EMP related features have been associated with aggressiveness and worse prognosis [2,3,4,5,6,7,8,9]. Similarly, IL-6 and OSM are linked to decreased overall and disease-free survival in various cancer types, in part due to their role in EMP and metastasis [147]. Here, we gather the literature-reported associations between IL-6, OSM and classical EMP markers in clinical samples.

Gallbladder cancer samples showed increased IL-6, twist, vimentin and decreased E-cadherin protein levels compared to the adjacent normal tissue, and the expression of all these markers significantly correlated with local invasion, lymph node status and patients’ clinical stage and inversely correlated with differentiation status [148].

Positive correlations between IL-6 and Snail, vimentin, Zeb1 and Zeb2 expression, and a negative association with miR-34a RNA levels were also reported in primary human CRC specimens [86]. Likewise, the expression of IL-6R negatively correlated with miR-34a and positively with Snail, Slug, vimentin, Zeb1, Zeb2 and p-STAT3 protein levels in those samples. Moreover, samples with positive lymph nodes or distant metastasis displayed increased levels of IL-6 and/or IL-6R and mesenchymal markers with a trend of decreased levels of miR-34a expression. All these data supported that the IL-6R/STAT3/miR-34a loop described in vitro is also active in primary human CRC and contributes to a mesenchymal phenotype and poor prognosis [86]. Additionally, immunohistochemistry (IHC) analysis in resected specimens of biliary cancer showed TGF-β and IL-6 staining not only in the invasive margins of the tumour but also in the invading cancer cells in the bile duct [60].

In lung adenocarcinoma clinical samples, the inverse correlation between E-cadherin and IL-6 expression, and the positive correlation between IL-6, vimentin and STAT3 protein phosphorylation confirmed the importance of the IL-6/STAT3/Snail axis described in animal models [79]. Another study in lung cancer reinforced those results, as it showed, by IHC analysis of the tumour specimens from patients with NSCLC after chemo(radio)therapy, that stromal IL-6 expression correlated with EMP changes in cancer cells (decreased E-cadherin and increased N-cadherin and TGFβ1 staining), suggesting that stromal IL-6 expression was an independent prognostic factor in patients with NSCLC [59].

More recently, tissue array analysis of 448 cancer samples from multiple organs revealed positive correlations between Zeb1 and IL-6 protein levels in various cancer types, including liver, colon and breast cancer [117]. Gene Set Enrichment Analysis (GSEA) of the TCGA-EAC dataset, in which oesophageal cancer samples were dichotomized by median IL-6 expression, revealed a significant association with two previously published EMT signatures. Additionally, low IL-6-expressing tumours are associated with an epithelial signature [90].

OSM has been proposed to be a marker of mesenchymal-like ER-negative/HER2-negative breast cancer, as high levels of OSM expression significantly correlated with several genes associated with EMP and EGF signalling, including IL6, NFKB1, CDH1, EGFR, MAP2K7, MAP3K2, MTOR, PIK3R2, TGFB1 and ZEB2 [149]. Moreover, OSM expression was found to be enriched in Sox2 or Snail-high tumours from an in-house cohort of human invasive breast carcinoma, and positively correlated with the mesenchymal protein S100A7 [48]. Regarding the OSM receptor OSMR, analysis of squamous cell carcinoma samples from TCGA demonstrated that OSMR expression positively correlated with all the analysed EMP-related genes (i.e., COL1A1, COL1A2, COL3A1, FN1, FOXC2, HMGA2, ITGA5, MMP10, MMP2, MMP9, SNAI1, SNAI2, TWIST2, VIM, ZEB1, ZEB2) in lung SCC, and with many of these genes in head and neck and cervical SCC [96]. Additionally, OSMR has been shown to significantly correlate with several EMP (SNAI1, SNAI2, ZEB1, ZEB2, TWIST1, CDH1, CDH2, VIM, FOXC1, FOXC2) and CSC (ABCG2, EGFR, NOTCH1, NOTCH4, SOX2, CD133, CD44/CD24, NANOG, POU5F1)-associated factors in fine needle aspirates of invasive breast carcinomas [48]. Moreover, OSMR expression exhibited a significant negative correlation with E-cadherin levels and a positive correlation with N-cadherin expression in gastric cancer samples from TCGA, and the results were validated by IHC analysis in an independent cohort [47].

In summary, the information gathered in this section supports that the association between IL-6, OSM and EMP observed in cancer cell lines and animal models is relevant in the clinical setting. All the information from clinical samples described above has been obtained from histopathological or gene expression analyses of tumour biopsies. Further evidence of the link between IL-6, OSM and EMP provided by more dynamic studies on primary human tumour explants or patient-derived xenografts (PDXs) would be desirable. 

## 3. Role of LIF on Epithelial-Mesenchymal Plasticity

The relevance of the leukaemia inhibitory factor (LIF) in stem cell maintenance and development is well described [150]. However, its role in cancer-related epithelial–mesenchymal plasticity is less clear, as both pro- and anti-EMP functions have been reported (Table 3, Figure 2).

In nasopharyngeal carcinoma, LIF reprograms the cancer cell invasive mode from collective to mesenchymal migration via acquisition of EMP, as shown by increased expression of N-cadherin, vimentin and IQ motif-containing GTPase-activating protein 1 (IQGAP1) concomitant with a decreased expression of E-cadherin [100]. In those cancer cells, LIF promoted invadopodia-associated characteristics and markers, such as tyrosine kinase substrate with five SH3 domains (TKS5), cortactin (CTTN), matrix metallopeptidase 2 (MMP2) and SRC proto-oncogene (SRC). Furthermore, LIF induced fibroblastic morphology with enhanced vascular dissemination and local invasion through modulation of the YAP1-FAK/PXN signalling in 3D gels and xenografts [100]. These results are relevant in the clinical setting, as higher levels of LIF and LIF receptor (LIFR) correlated with poorer metastasis/recurrence-free survival [100].

Similar in vitro morphological changes, accompanied by increased vimentin and N-cadherin, and decreased E-cadherin mRNA and protein levels, have been reported to be activated by LIF in the human CRC cell line HCT116 and in epithelial-like T47D and MCF7 breast cancer cell lines in vitro [101]. These LIF effects were shown to be mediated by STAT3-induced miR-21 overexpression [101]. In chondroma cells, LIF treatment increased migration, invasion, chemoresistance, colony and tumourosphere formation and regulated the expression of EMP and the stemness markers (increased Zeb2, CD15 and CD133 and reduced E-cadherin and CK-19) [103]. However, these effects were not accompanied by any morphological changes [103].

Other studies have also associated LIF signalling with EMP features such as increased cell migration, invasion, CSC properties and metastasis, reporting both promoting and inhibitory roles, despite not always showing a direct link between LIF and EMP. In gastric cancer, LIF promoted proliferation, colony formation, invasion, migration and tumour growth by inhibiting the Hippo pathway, resulting in increased YAP nuclear translocation and transcriptional activity, giving rise to cancer cells with decreased E-cadherin and increased MMP7 protein levels [104]. Interestingly, LIF promoted mesenchymal epithelial transition in lung cancer cells through the LIF/LIFR/p-ERK/pS727-STAT3 signalling pathway leading to increased metastasis [42]. 

Intriguingly, in breast cancer, anti-metastatic potential has been attributed to LIFR [102,151], as LIFR downregulation was responsible for the pro-metastatic effect of the E-cadherin suppressor miR-9 [102]. Additionally, in the TNBC cell line MDA-MB-231, LIFR overexpression did not alter vimentin protein levels in vitro but reduced vimentin positive metastatic foci in vivo [102]. The anti-tumorigenic roles of LIF in gastric CSCs also have been described, where it decreased the CSC properties and population in both cell lines and a patient-derived xenograft (PDX) [105] (Figure 2). Both in breast and in gastric cancer cells, the anti-metastatic role of LIF-LIFR was mediated through the activation of Hippo tumour suppressor kinases (MST1/2 and LATS1/2), with the consequent inhibition of the YAP/TAZ/TEAD oncogenic effector activity. It would be interesting to know if STAT1 is involved in the anti-metastatic effect of LIF, as has been described for other cytokines of the family [69].

## 4. Role of IL-11 on Epithelial–Mesenchymal Plasticity

Interleukin-11 (IL-11) is known to participate in osteoclast-mediated bone remodelling together with TGFβ, and both have been associated with increased bone metastasis [152,153]. However, reports linking IL-11 and EMP are limited, as far as we know, with few exceptions supporting a positive contribution of IL-11 to EMP features. These results are described below and summarized in Table 3 and Figure 2.

IL-11 is increased by HIF1α in hypoxia and induces EMP via the PI3K/Akt/GSKβ3/Snai1 pathway in anaplastic thyroid carcinoma cells, as demonstrated by reduced ZO-1 and E-cadherin and increased vimentin and Snail protein levels, ultimately improving their migratory and invasive potential [106]. IL-11 treatment also promoted EMP in different in vivo NSCLC models, via AKT and STAT3, leading to increased levels of mesenchymal markers, such as Snail, Slug, Twist1, vimentin and N-cadherin, and downregulation of E-cadherin, claudin-1 and ZO-1 expression [107,108]. One of these studies reported that IL-11 treatment promoted lung adenocarcinoma cell growth and EMT through activation of the STAT3/HIF-1α/EMT signalling pathway [107]. In TNBC cells, IL-11 cooperated with its upstream regulator twinfilin 1 in promoting EMT and chemoresistance. In addition, knockdown of IL-11 favoured a mesenchymal-to-epithelial transition phenotype as demonstrated by rearranged actin filaments and increased vinculin-stained focal adhesions [109]. The relevance in the clinical setting was demonstrated as low IL-11 correlated with relapse-free survival in breast cancer patients [109]. Additionally, the IL-11/STAT3 pathway, activated by the oncogene HMGA2, has been shown to facilitate the migratory and invasive capacities of colorectal cells in vitro together with increased vimentin and decreased E-cadherin levels, and to promote tumorigenesis and distant metastasis in vivo [110]. Of interest, the oncogene HMGA2 is also a master regulator of OSM-induced epithelial plasticity in breast cancer and can be induced by OSM [97], as we described in previous sections.

A recent study has shown that, in PDAC cells, IL-6 and IL-11 stimulated the expression of most S100 proteins regulating epithelial/mesenchymal features [89]. Even if IL-11 treatment alone did not induce EMP by itself in this study, it synergised with IL-6 to activate STAT3, which cooperated with ZEB1 to upregulate the mesenchymal S100A4/A6 proteins, nullify the effect of epithelial S100A14 expression and promote an invasive phenotype [89].

In conclusion, the available literature supports the role of IL-11 as a pro-EMP and pro-tumorigenic factor in anaplastic thyroid, lung, breast, colorectal and pancreatic cancer, mainly through activation of the STAT3 and PI3K/Akt pathways [89,106,107,108,109,110].

## 5. Role of IL-27 and IL-30 on Epithelial–Mesenchymal Plasticity

Interleukin-27 (IL-27), which also belongs to the IL-12 family, has been described as a potential anti-tumour therapeutic agent [154,155]. Surprisingly, independent pro-tumorigenic effects have been attributed to the IL-27/p28 subunit or IL-30 cytokine in prostate and breast cancers, which have been recently reviewed [156,157]. However, there is scarce information about any direct link between IL-27 and IL-30 with EMP as far as we know.

IL-27 strongly protects the epithelial phenotype by a dominant STAT1 pathway, even in the presence of TGFβ, by cooperating with SMADs in lung cancer cells [112]. IL-27 decreased Snail, Slug, Zeb1, vimentin and N-cadherin expression; increased the E-cadherin, β-catenin and γ-catenin levels; and reduced the migration, secretion of angiogenic factors (IL-8, VEGF, CXCL5) and markers of stemness (SHH, OCT4A, SOX2, SOX9, NOTCH1, KLF4, nestin), both in vitro and in vivo [111,112,113] (Table 3 and Figure 2). 

Although direct evidence between IL-30 and the classical EMP markers remain elusive, IL-30 was shown to induce TNBC cell migration and the expression of a pro-oncogenic program to promote breast cancer growth and progression via STAT3 [158]. Even if in this study most of the analysed EMP-TFs remained substantially unchanged after IL-30 treatment, IL-6 and Sonic Hedgehog (SHH) were consistently upregulated both in vitro and in vivo [158]. Regarding prostate cancer, IL-30 expression correlates with advanced disease grade and stage [159], and it has been shown to play an important role in regulating prostate CSC-like behaviour and metastatic potential [160]. 

These opposite effects between IL-27 and IL-30 reinforce the importance of the STAT1/STAT3 balance in tuning the effect of IL-6 cytokines on EMP, STAT3 being a potent EMP inducer and STAT1 an anti-EMP regulator [161].

## 6. Conclusions

IL-6 family cytokines share many similarities as they have the potential to activate the same signalling pathways (e.g., JAK/STATs, MAPK/ERK and PI3K/AKT), but play different roles and are involved in different physiological and pathological processes as it is evidenced in the case of EMP. IL-6 and OSM are clear EMP promoters and their role in tumour progression and metastasis is widely accepted (Figure 1). IL-11 and LIF seem to promote EMP in some cancer types, even if the mechanisms are not as thoroughly described (Figure 2). However, the implication of LIF is less clear as it can exert both pro and anti-metastatic functions and their link with EMP is less studied. IL-27 is the unique strong EMP inhibitor of the family by a dominant STAT1 activation, while, interestingly, its subunit p28 (IL-30) has been suggested to favour an invasive phenotype in breast cancer cells (Figure 2).

Although it is widely accepted that IL-6 and OSM promote EMP in many cancer types, the data gathered in this review further suggest that both promote different EMP programmes depending on the cancer type and the experimental setting, giving rise to cells with intermediate phenotypes within the epithelium–mesenchymal spectrum rather than generating complete mesenchymal cells with no epithelial features. Repression of E-cadherin and increased Snail levels seem to be common features of the EMP programmes activated by IL-6 and OSM, while there is more discrepancy in the mesenchymal markers and other core EMT-TFs induced by those cytokines. The tumorigenic properties of IL-6 and OSM are principally mediated by STAT3, while few anti-EMP functions of OSM have been described by a dominant STAT1 activation in specific contexts. 

Even if the EMP process induced by IL-6 and OSM has been shown to be reversible, there are several autocrine and paracrine positive feedback loops, leading to long-term pro-EMP signalling. In addition, both cytokines are required for the initiation and maintenance of transdifferentiated/mesenchymal and CSC-like populations, supporting a role in the acquisition of therapy resistance [44,55,77,134,147,162,163]. In conclusion, the data presented in this review, obtained from in vitro and in vivo experiments and from clinical samples, strongly supports the role of IL-6 and OSM on EMP promotion and suggests that both cytokines could be potential therapeutic targets to halt tumour progression by blocking EMP.

## Figures and Tables

**Figure 1 ijms-22-08334-f001:**
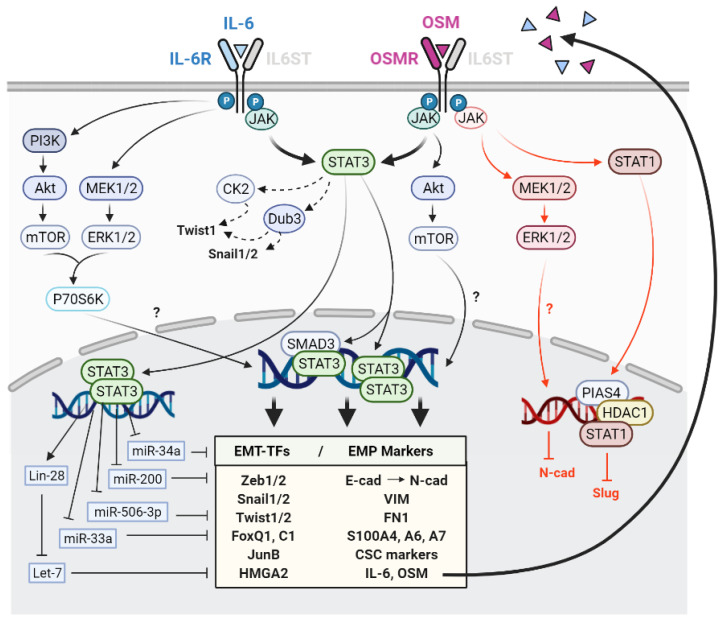
Schematic representation of the effects of interleukin 6 (IL-6) and oncostatin M (OSM) in epithelial–mesenchymal plasticity (EMP) and the downstream pathways and mediators involved. Black and red colours indicate pro and anti-EMP functions, respectively. T arrows indicate inhibition. Dashed arrows indicate EMP promotion by EMT-TF protein stabilisation. EMT-TFs: Transcription factors involved in EMP.

**Figure 2 ijms-22-08334-f002:**
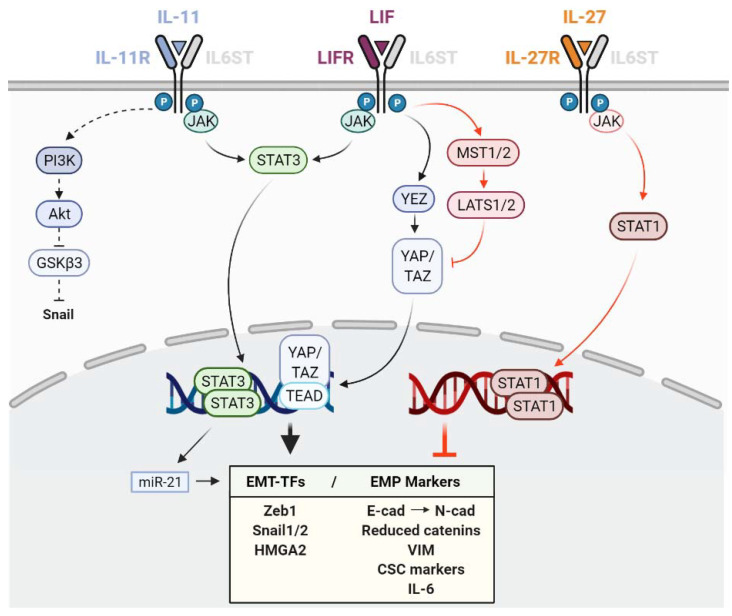
Schematic representation of the different effects of interleukin 11 (IL-11), leukaemia inhibitory factor (LIF) and interleukin 27 (IL-27) in epithelial–mesenchymal plasticity (EMP) and the downstream pathways and mediators involved. Black and red colours indicate pro- and anti-EMP functions, respectively. T arrows indicate inhibition. Dashed arrows indicate EMP promotion by EMT-TF protein stabilization. EMT-TFs: Transcription factors involved in EMP.

**Table 1 ijms-22-08334-t001:** Experimental evidence of the implication of interleukin 6 (IL-6) on epithelial–mesenchymal plasticity (EMP) in cancer.

Cancer Type	Main Pathway	EMT-TFs	EMT Markers	References
Breast	STAT3	Snail Twist1 Zeb1	Mesenchymal-like morphology Decreased E-cadherin Increased N-cadherin, vimentin, CSC markers (CD24low/CD44high)	Walter 2009 [74]; Sullivan 2009 [72]; Gyamfi 2018 [73]; Y Wu 2017 [75]; Xie 2012 [76]; Korkaya 2012 [77]; Marotta 2011 [78]
Lung	STAT3	Snail Slug Twist1	Mesenchymal-like morphology Decreased E-cadherin Increased vimentin, nuclear β-catenin, CSC markers (CD133)	Zhao 2014 [79]; Shintani 2016 [59]; Lee 2016 [80]; Liu 2020 [65]; Che 2017 [81]
Head and neck	STAT3	Snail Twist1	Decreased E-cadherin Increased vimentin	Son 2015 [82]; Yadav 2011 [45]; Su 2011 [83]
Colon	STAT3	Snail Slug Zeb1 TCF3 Fra-1 FoxQ1	Mesenchymal-like morphology Decreased E-cadherin, ZO-1 Increased N-cadherin, vimentin, fibronectin, β-catenin, cell scattering	Liu 2015 [84]; Wei 2019 [85]; Rokavec 2014 [86]; Gao 2018 [58]
Pancreas	STAT3	Snail Slug Zeb2 Twist2 Nrf2	Mesenchymal-like morphology Decreased E-cadherin Increased N-cadherin, vimentin, fibronectin, collagen I, CSC markers (CD133), S100A4, S100A6, MMP9	YS Wu 2017 [87]; Kesh 2020 [88]; Al-Ismaeel 2019 [89]; Li 2020 [66]
Oesophagus	STAT3	Zeb1 Slug	Mesenchymal-like morphology Decreased E-cadherin, CK19, EPCAM Increased N-cadherin, vimentin, CSC markers (CD24low/CD44high, CD133)	Ebbing 2019 [90]
Bladder	Unknown	Snail Twist1 Zeb1	Decreased E-cadherin, β-catenin Increased N-cadherin, vimentin	Goulet 2019 [91]
Gallbladder	Unknown	Twist1	Increased CSC markers (CD44, CD133)	Zhang 2016 [92]
Uveal melanoma	STAT3	JunB	Decreased E-cadherin, TJP2, TJP2 Increased fibronectin and fibronectin receptors, ICAM1	Gong 2018 [93]
Cervix	STAT3	Unknown	Mesenchymal-like morphology Decreased E-cadherin Increased vimentin, cell scattering	Miao 2014 [94]

**Table 2 ijms-22-08334-t002:** Experimental evidence of the implication of oncostatin M (OSM) on epithelial–mesenchymal plasticity (EMP) in cancer.

Cancer Type	Main Pathway	EMT-TFs	EMT Markers	References
Pancreas	STAT3	Snail Zeb1	Mesenchymal-like morphology Decreased E-cadherin CSC markers (CD24low/CD44high)	Smigiel 2017 [95]
Cervix	STAT3	Snail Zeb2	Mesenchymal-like morphology Reduced E-cadherin, cell cohesion Increased vimentin, fibronectin, MMP9, MMP10	Kucia-Tran 2016 [96]
Breast	STAT3 and PI3K/AKT	Snail Slug Zeb1 FoxC1	Mesenchymal-like morphology Decreased E-cadherin, claudin-1, β-catenin Increased N-cadherin, vimentin, fibronectin, MMPs, S100A7, CSC markers (CD24low/CD44 high)	Doherty 2019 [61]; Guo 2013 [97]; Junk 2017 [43]; West 2014 [48]; Lapeire 2014 [98]; Bryson 2017 [49]; Parashar 2019 [99]
Prostate	STAT3	Snail	Mesenchymal-like morphology, Decreased E-cadherin Increased vimentin	Sterbova 2018 [51]
Stomach	STAT3		Decreased E-cadherin Increased N-cadherin	Yu 2019 [47]
Lung	STAT1	Slug	Increased E-cadherin	Pan 2016 [69]
Kidney	STAT3		Decreased E-cadherin, claudin-2 Increased vimentin, collagen I, S100A4, cell scattering	Pollack 2007 [71]
	ERK1/2		Decreased N-cadherin	Pollack 2007 [71]
	STAT1	FoxC2	Decreased secretion of matricellular proteins (SPARC, thrombospondin, CTGF, TNC)	Sarközi 2011 [70]

**Table 3 ijms-22-08334-t003:** Experimental evidence of the implication of leukaemia inhibitory factor (LIF), interleukin 11 (IL-11) and 27 (IL-27) on epithelial–mesenchymal plasticity (EMP) in cancer.

Cytokine	Cancer Type	Main Pathway	EMT-TFs	EMT Markers	References
LIF	Head and Neck	Hippo pathway (YAP/FAK/PXN)		Mesenchymal-like morphology Decreased E-cadherin Increased N-cadherin, vimentin, IQAP1	Liu 2018 [100]
LIF	Colon	STAT3		Mesenchymal-like morphology Decreased E-cadherin Increased N-cadherin, vimentin	Yue 2016 [101]
LIF	Breast	STAT3		Mesenchymal-like morphology Decreased E-cadherin Increased N-cadherin, vimentin	Yue 2016 [101]
		STAT3/ Hippo (MST/LATS/YAP)		Decreased YAP	Chen 2012 [102]
LIF	Chondroma		Zeb2	Decreased E-cadherin, CK19, Increased CSC markers (CD15, CD133)	Gulluoglu 2017 [103]
IL-11	Stomach	Hippo pathway (Hippo/ YAP)		Decreased E-cadherin Increased YAP, MMP7	Bian 2020 [104]
		STAT3/ Hippo (MST/LATS/YAP)		Decreased YAP and stemness	Seeneevassen 2020 [105]
IL-11	Thyroid	PI3K/AKT	Snail	Decreased E-cadherin, ZO-1 Increased vimentin	Zhong 2016 [106]
IL-11	Lung	STAT3	Snail Slug Twist1	Decreased E-cadherin, ZO-1, claudin-1 Increased N-cadherin, vimentin	Zhao 2018 [107] Peng 2020 [108]
IL-11	Breast			Mesenchymal-like morphology F-actin organization, increased focal adhesions	Bockhorn 2013 [109]
IL-11	Colon	STAT3	HMGA2	Increased vimentin	Wu 2016 [110]
IL-11	Pancreas	STAT3		Mesenchymal-like morphology Increased S100A4, S100A6	Ai-Ismaeel 2019 [89]
IL-27	Lung	STAT1	Snail Slug Zeb1	Increased E-cadherin, β-catenin, γ-catenin Reduced N-cadherin, vimentin, CSC markers (SHH, OCT4A, SOX2, SOX9, NOTCH1, KLF4, nestin)	Airoldi 2015 [111]; Dong 2016 [112]; Kachroo 2013 [113]

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
