# Peer review of "The Role of the IL-6 Cytokine Family in Epithelial–Mesenchymal Plasticity in Cancer Progression"

_ijms, 2021, doi:10.3390/ijms22158334_

Round 1

Reviewer 1 Report

Comprehensive summary on the cytokine IL-6 family and its relation to EMP in cancer. Authors provide rather detailed account of the published evidence related to the mostly positive role of IL-6 family members in expansive behavior of tumor cells of several types of solid malignancies. Basically, the style and form of the provided information is no different from other similar reviews; i.e. a series of documented facts obtained mostly from various tumor cell lines and providing minute detailes over mechanistic aspects of the discussed problematic. In the reviewer´s opinion the most valuable part of this review relates to the clinical evidence; that is correlation between experimental evidence in vitro and histopathological findings from actual tumor biopsies. Sadly, no evidence (at least not emphasized and discussed) is provided from primary tumor explants which would ideally bridge the evidence from stabilized lines and "static detections" originating from biopsies. Also, sadly enough, this review suffers from the common drawback of present days summaries. Too much focus on details with merely listing series of papers rather than critical evaluation of the published findings in the context of the expected message. The interested reader is thus often swept by particular minutiae and loses the overall view. Here it is imperative that authors truly discuss available evidence and not just list and repeat the findings of others. It is in particular important in instances where "conflicting" findings exist which cannot be just dismissed by simple stating that "these findings arose due to diferent experimental set up". 

Minor comment:

in some cases inappropriate words are used: pp.3 ..."cannot be dicarded" - should be cannot be disregarded; "cells suffered EMP" - should be cells underwent EMP"

pp. 5 - Fig. 1 legend - discontinuous line - should be dashed line

In the introduction it is stated that EMT and MET are important for embryogenesis and cancer which is not complete statement as related to the following text. Authors should state that these are important both for physiological processes (embryogenesis, regeneration) as well as pathgologiical processes (cancer, fibrosis).

Author Response

Response to Referee 1

Comprehensive summary on the cytokine IL-6 family and its relation to EMP in cancer. Authors provide rather detailed account of the published evidence related to the mostly positive role of IL-6 family members in expansive behavior of tumor cells of several types of solid malignancies. Basically, the style and form of the provided information is no different from other similar reviews; i.e. a series of documented facts obtained mostly from various tumor cell lines and providing minute detailes over mechanistic aspects of the discussed problematic. In the reviewer´s opinion the most valuable part of this review relates to the clinical evidence; that is correlation between experimental evidence in vitro and histopathological findings from actual tumor biopsies. Sadly, no evidence (at least not emphasized and discussed) is provided from primary tumor explants which would ideally bridge the evidence from stabilized lines and "static detections" originating from biopsies.

We thank the reviewer for the helpful comments. Unfortunately, there is no evidence about the role of IL-6 related cytokines on EMT originated from primary tumor explants or patient derived xenografts (PDXs). We have included a sentence in page 11 to highlight the need for these studies. We have also highlighted the evidence obtained from in vivo mouse cancer models and not only from in vitro experiments with cancer cell lines (pages 7, 9, 11, and 15).

Also, sadly enough, this review suffers from the common drawback of present days summaries. Too much focus on details with merely listing series of papers rather than critical evaluation of the published findings in the context of the expected message. The interested reader is thus often swept by particular minutiae and loses the overall view. Here it is imperative that authors truly discuss available evidence and not just list and repeat the findings of others. It is in particular important in instances where "conflicting" findings exist which cannot be just dismissed by simple stating that "these findings arose due to diferent experimental set up".

We agree with the reviewer and, in order to improve the manuscript and strengthen the discussion, we have included sentences in each section summarising and discussing the main results and highlighting the most important findings. Likewise, in the case of conflicting findings, we have discussed those results in the context of the available evidence. In some cases, we have deleted the references to small details observed only in one single cell study when most of the literature pointed to another direction. Please, see new sentences/paragraphs on pages 4, 6, 7, 8, 9, 11, 12, 13, 14 and 15.

Minor comment:

in some cases inappropriate words are used: pp.3 ..."cannot be dicarded" - should be cannot be disregarded; "cells suffered EMP" - should be cells underwent EMP"

These errors have been corrected.

  1. 5 - Fig. 1 legend - discontinuous line - should be dashed line

This has been corrected for Figure 1 and 2 legends.

In the introduction it is stated that EMT and MET are important for embryogenesis and cancer which is not complete statement as related to the following text. Authors should state that these are important both for physiological processes (embryogenesis, regeneration) as well as pathgologiical processes (cancer, fibrosis).

This sentence has been rephrased in the introduction, page 1.

Reviewer 2 Report

This is an excellent and timely review! Very comprehensive as well. 

Just a couple of minor suggestions:

1-EMP/EMT confers resistance to anoikis--discuss?  mention?

2- You might want to mention something about the regulation of EMT-suppressive factors like Grainyhead-like-2 , to complement your thorough discussion of EMT-promoting factors.

Author Response

Response to Referee 2

This is an excellent and timely review! Very comprehensive as well.

Just a couple of minor suggestions:

1-EMP/EMT confers resistance to anoikis--discuss?  mention?

We thank the reviewer for his/her comments. We have now included a mention to the effect of IL-6 on resistance to anoikis (page 7). To our knowledge, there is no experimental evidence on the link of OSM, LIF, IL-11, IL-27 and IL-30 to anoikis resistance.

2- You might want to mention something about the regulation of EMT-suppressive factors like Grainyhead-like-2 , to complement your thorough discussion of EMT-promoting factors.

We thank the reviewer for the suggestion. However, we have not found any report on the regulation of EMT suppressive factors like Grainyhead-like-2 (GRHL2), OVOL1/2, ELF3, ELF5 by cytokines of the IL-6 family. Therefore, we have not included any mention to these suppressive factors in the manuscript.
